# Antigenicity of the Mu (B.1.621) and A.2.5 SARS-CoV-2 Spikes

**DOI:** 10.3390/v14010144

**Published:** 2022-01-14

**Authors:** Debashree Chatterjee, Alexandra Tauzin, Annemarie Laumaea, Shang Yu Gong, Yuxia Bo, Aurélie Guilbault, Guillaume Goyette, Catherine Bourassa, Gabrielle Gendron-Lepage, Halima Medjahed, Jonathan Richard, Sandrine Moreira, Marceline Côté, Andrés Finzi

**Affiliations:** 1Centre de Recherche du CHUM, Montreal, QC H2X 0A9, Canada; debashree.chatterjee@umontreal.ca (D.C.); alexandra_tauzin@hotmail.fr (A.T.); annemarie.laumaea@umontreal.ca (A.L.); shang.gong@mail.mcgill.ca (S.Y.G.); guillaume@biotechconnect.ca (G.G.); catherine.bourassa.chum@ssss.gouv.qc.ca (C.B.); gabrielle.gendron-lepage.chum@ssss.gouv.qc.ca (G.G.-L.); halima.medjahed.chum@ssss.gouv.qc.ca (H.M.); jonathan.richard.1@umontreal.ca (J.R.); 2Département de Microbiologie, Infectiologie et Immunologie, Université de Montréal, Montreal, QC H2X 0A9, Canada; 3Department of Microbiology and Immunology, McGill University, Montreal, QC H3A 0G4, Canada; 4Department of Biochemistry, Microbiology and Immunology, and Center for Infection, Immunity, and Inflammation, University of Ottawa, Ottawa, ON K1H 8M5, Canada; byuxia@uottawa.ca (Y.B.); Marceline.Cote@uottawa.ca (M.C.); 5Laboratoire de Santé Publique du Québec, Institut Nationale de Santé Publique du Québec, Sainte-Anne-de-Bellevue, QC H9X 3R5, Canada; aurelie.guilbault@inspq.qc.ca (A.G.); sandrine.moreira@inspq.qc.ca (S.M.)

**Keywords:** coronavirus, COVID-19, SARS-CoV-2, spike glycoproteins, RBD, ACE2, temperature, variants of concern, variants of interest, variants under monitoring, mRNA vaccines

## Abstract

The rapid emergence of SARS-CoV-2 variants is fueling the recent waves of the COVID-19 pandemic. Here, we assessed ACE2 binding and antigenicity of Mu (B.1.621) and A.2.5 Spikes. Both these variants carry some mutations shared by other emerging variants. Some of the pivotal mutations such as N501Y and E484K in the receptor-binding domain (RBD) detected in B.1.1.7 (Alpha), B.1.351 (Beta) and P.1 (Gamma) are now present within the Mu variant. Similarly, the L452R mutation of B.1.617.2 (Delta) variant is present in A.2.5. In this study, we observed that these Spike variants bound better to the ACE2 receptor in a temperature-dependent manner. Pseudoviral particles bearing the Spike of Mu were similarly neutralized by plasma from vaccinated individuals than those carrying the Beta (B.1.351) and Delta (B.1.617.2) Spikes. Altogether, our results indicate the importance of measuring critical parameters such as ACE2 interaction, plasma recognition and neutralization ability of each emerging variant.

## 1. Introduction

Severe acute respiratory syndrome coronavirus 2 (SARS-CoV-2), the etiological agent of COVID-19, infected more than 300 million individuals and caused over 5.5 million deaths [1]. The Spike glycoprotein (S) mediates viral entry. S is composed of two subunits (S1 and S2): S1 contains the receptor binding domain (RBD) responsible for interaction with the receptor angiotensin-converting enzyme 2 (ACE2) expressed on host cells [2,3,4], whereas S2 domain facilitates the fusion of viral–host membranes [5,6]. The Spike glycoprotein is a major target of cellular and humoral responses elicited by natural infection. Accordingly, currently approved adenoviral vectors and mRNA vaccine platforms target the Spike [7,8,9,10].

While the presence of proofreading exonucleases limits the rate of misincorporation in coronaviruses compared to many other RNA viruses [11], SARS-CoV-2 variants still emerge with unique properties. Indeed, recombination, insertion and deletion events in the viral genome of this virus lead to the emergence of new variants with enhanced transmission capabilities and resistance to neutralizing antibodies elicited by natural infection and vaccination. These variants are classified as variants of concern (VOCs), variants of interest (VOIs) or variants under monitoring (VUMs), according to the World Health Organization (WHO) [1]. In late 2020, several VOCs emerged named B.1.1.7 (Alpha), B.1.351 (Beta), P.1 (Gamma) as well as the VOIs B.1.429 (Epsilon), B.1.526 (Iota), B.1.617.1 (Kappa) and B.1.617 [12,13,14,15,16,17,18]. Lota and Kappa are now classified as VUMs [1]. In April 2021, the Delta variant (B.1.617.2) became prevalent in India and rapidly spread all over the world [12,19].

Recently, the Mu (B.1.621) variant, first identified in Colombia during March–April 2021, was classified as a variant of interest [12]. The Spike of Mu accumulated the following mutations: insertion in 146N, T95I, Y144T and Y145S, in the N-terminal domain (NTD); R346K, E484K, N501Y in the receptor-binding domain (RBD) and P681H at the S1/S2 interface [20]. In Québec (Canada), an increase of the number of cases with the Mu variant during the summer (Appendix A) and a relative slowdown of Delta prevalence compared to the progression observed in other countries raised the question of a possible evolutionary advantage for this variant.

Another variant, A.2.5, likely spread north from Central America to the Quebec province in Canada and accumulated deletions 141–143 at the NTD and the L452R mutations in RBD. This variant belongs to the A.* lineages, which predominated early during the pandemic but were replaced by B lineages characterized by the D614G mutation on the Spike glycoprotein until becoming quasi-extinct in mid-2020. The resurgence of A.* lineages was observed in different countries at the beginning of 2021. In Québec, the lineage A.2.5 was detected in two outbreaks with high secondary attack rate. Further analysis revealed the convergent acquisition of the D614G and L452R signature mutations of the B lineages and Delta variant, respectively, and a mutational jump of 23 mutations, a characteristic of most of the VOCs [21]. These observations raised the hypothesis of a fitness advantage of the A.2.5 clade.

To gain a better understanding of the antigenic properties of Mu and A.2.5 Spikes, we evaluated their capacity to interact with ACE2 and performed binding and neutralization assays with plasma from vaccinated individuals. Since the mutations of these Spikes are close to those defining the Beta and Delta lineages, we used Spikes from these VOCs as controls.

## 2. Material and Methods

### 2.1. Ethics Statement

All work was conducted in accordance with the Declaration of Helsinki in terms of informed consent and approval by an appropriate institutional board. Blood samples were obtained from donors who consented to participate in this research project at Centre de Recherche du CHUM and approved by the CHUM Research Ethics Board (No. 19.381).

### 2.2. Plasmas and Antibodies

Plasmas were isolated by centrifugation with Ficoll gradient and stored at −80 °C or for further use. Collected plasmas were heat-inactivated for 1 h at 56 °C and stored at −80 °C until use in subsequent experiments. Healthy donors’ plasmas, collected before the pandemic, were used as negative controls in flow cytometry and neutralization assays. The conformationally independent S2-specific monoclonal antibody CV3-25 [22,23] was used as a positive control and to normalize Spike expression in our flow cytometry assays, as described [22,24,25]. ACE2 binding was measured using the recombinant ACE2-Fc protein, which is composed of two ACE2 ectodomains linked to the Fc portion of the human IgG [26]. Alexa Fluor-647-conjugated goat anti-human Abs (Invitrogen, Waltham, MA, USA) were used as secondary antibodies to detect ACE2-Fc and plasma binding in flow cytometry experiments.

### 2.3. Protein Expression and Purification

FreeStyle 293 F cells (Invitrogen) were grown in FreeStyle 293F medium (Invitrogen) to a density of 1 × 10^6^ cells/mL at 37 °C with 8% CO_2_ with regular agitation (150 rpm). Cells were transfected with a plasmid coding for SARS-CoV-2 S RBD using ExpiFectamine 293 transfection reagent, as directed by the manufacturer (Invitrogen). One week later, cells were pelleted and discarded. Supernatants were filtered using a 0.22 μm filter (Thermo Fisher Scientific, Waltham, MA, USA). The recombinant RBD proteins were purified by nickel affinity columns, as directed by the manufacturer (Invitrogen). The RBD preparations were dialyzed against phosphate-buffered saline (PBS) and stored in aliquots at −80 °C until further use. To assess purity, recombinant proteins were loaded on SDSPAGE gels and stained with Coomassie Blue.

### 2.4. Cell Lines

293T human embryonic kidney cells (obtained from ATCC) were maintained at 37 °C under 5% CO_2_ in Dulbecco’s modified Eagle’s medium (DMEM) (Wisent, Saint-Jean-Baptiste, QC, Canada) containing 5% fetal bovine serum (VWR) and 100 μg/mL of penicillin-streptomycin (Wisent). The 293T-ACE2 cell line was previously described [27] and was maintained in medium supplemented with 2 µg/mL of puromycin (Millipore Sigma, Burlington, MA, USA).

### 2.5. Plasmids

The plasmids encoding B.1.351 Spike was codon-optimized and synthesized by Genscript. Plasmids encoding B.1.617.2, B.1.621 and A.2.5 Spikes were generated by overlapping PCR using a codon-optimized wild-type SARS-CoV-2 Spike gene (GeneArt, ThermoFisher, Waltham, MA, USA) that was synthesized (Biobasic, Markham, ON, Canada) and cloned in pCAGGS as a template. The pCG1-SARS-CoV-2 S was kindly provided by Stefan Pöhlmann, its D614G variant was previously described [24,28]. The plasmid encoding for the recombinant RBD (residues 319–541) fused with a hexa-histidine tag was previously reported [24,28]. The presence of the desired mutations was determined by automated DNA sequencing.

### 2.6. Virus Neutralization Assay

293T cells were transfected with the lentiviral vector pNL4.3 R-E- Luc (NIH AIDS Reagent Program) and a plasmid encoding for the indicated Spike glycoprotein at a ratio of 10:1. Two days post-transfection, cell supernatants were harvested and stored at −80 °C until use. 293T-ACE2 target cells were seeded at a density of 1 × 10^4^ cells/well in 96-well luminometer-compatible tissue culture plates (Perkin Elmer, Waltham, MA, USA) 24h before infection. Pseudoviral particles were incubated with the indicated plasma dilutions (1/50; 1/250; 1/1250; 1/6250; 1/31250) for 1 h at 37 °C and were then added to the target cells followed by incubation for 48 h at 37 °C. Then, cells were lysed by the addition of 30 µL of passive lysis buffer (Promega, Madison, WI, USA) followed by one freeze-thaw cycle. A LB942 TriStar luminometer (Berthold Technologies, Bad Wildbad, Germany) was used to measure the luciferase activity of each well after the addition of 100 µL of luciferin buffer (15 mM MgSO_4_, 15 mM KPO_4_ [pH 7.8], 1 mM ATP, and 1 mM dithiothreitol) and 50 µL of 1 mM d-luciferin potassium salt (Prolume, Randolph, VT, USA). The neutralization half-maximal inhibitory dilution (ID_50_) represents the plasma dilution to inhibit 50% of the infection of 293T-ACE2 cells by SARS-CoV-2 pseudoviruses.

### 2.7. Cell Surface Staining and Flow Cytometry Analysis

293T were transfected with full length SARS-CoV-2 Spikes and a green fluorescent protein (GFP) expressor (pIRES2-eGFP; Clontech), CA, USA using the calcium–phosphate method. Two days post-transfection, 293T-Spike cells were stained with the CV3-25 Ab as control, ACE2-Fc or plasma from vaccinated individuals. Briefly, 5 µg/mL CV3-25 or 20 µg/mL ACE2-Fc were incubated with cells at 37 °C or 4 °C for 45 min. Plasma from above mentioned donors were incubated with cells at 37 °C. AlexaFluor-647-conjugated goat anti-human IgG (1/1000 dilution) were used as secondary Abs. The percentage of Spike-expressing cells (GFP+ cells) was determined by gating the living cell population based on viability dye staining (Aqua Vivid, Invitrogen, Waltham, MA, USA). Samples were acquired on a LSRII cytometer (BD Biosciences, Franklin Lakes, NJ, USA), and data analysis was performed using FlowJo v10.7.1 (Tree Star). CV3-25 was used to normalize Spike expression, as reported [24]. The Median Fluorescence intensities (MFI) obtained with ACE2-Fc or plasma Abs were normalized to the MFI obtained with the conformationally and temperature independent CV3-25 anti-S2 antibody [22,23,29] and presented as ratio of the CV3-25-normalized values obtained with the D614G Spike.

### 2.8. Biolayer Interferometry

Binding kinetics were performed on an Octet RED96e system (ForteBio, Fremont, CA, USA) at 25 °C and 10 °C with shaking at 1000 RPM. Amine Reactive Second-Generation (AR2G) biosensors were hydrated in water, then activated for 300 s with an S–NHS/EDC solution (ForteBio) prior to amine coupling. SARS-CoV-2 RBD proteins were loaded into AR2G biosensor at 12.5 μg/mL in 10 mM acetate solution pH5 (ForteBio) for 600 s and then quenched into 1M ethanolamine solution pH8.5 (ForteBio) for 300 s. Baseline equilibration was collected for 120 s in 10X kinetics buffer. Association of sACE2 (in 10X kinetics buffer) to the different RBD proteins was carried out for 180 s at various concentrations in a two-fold dilution series from 500 nM to 31.25 nM prior to dissociation for 300 s. The data were baseline subtracted prior to fitting performed using a 1:1 binding model and the ForteBio data analysis software. Calculation of on-rates (Ka), off-rates (Kdis), and affinity constants (K_D_) was computed using a global fit applied to all data.

## 3. Results

### 3.1. Impact of Temperature on ACE2 Binding by SARS-CoV-2 Spike

Since the Spikes of some emerging variants were reported to interact more efficiently with the ACE2 receptor, we measured the capacity of Mu and A.2.5 binding ability to recombinant ACE2-Fc protein [24] and compared to the Beta and Delta Spikes. Plasmids expressing the different SARS-CoV-2 full Spikes were transfected into HEK 293T cells. Spike expression was normalized with the conformationally independent, S2-specific CV3-25 monoclonal antibody, as described [22,23,24,29]. All Spikes tested bound significantly better to ACE2 compared to their D614G counterpart, with Beta presenting the higher binding (Figure 1A and Appendix A).

A recent study showed that temperature affects the capacity of SARS-CoV-2 S to interact with ACE2 [22]. Interestingly, we showed that this property was conserved among several SARS-CoV-2 emerging variants [24]. To evaluate whether temperature also affect the capacity of Mu and A.2.5 Spikes to interact with ACE2, we incubated Spike-expressing cells at either 37 °C or 4 °C for 45 min and measured ACE2-Fc binding by flow cytometry. As presented in Figure 1B, 4 °C incubation enhanced ACE2 interaction for all Spikes.

The impact of cold temperature (4 °C) on ACE2 binding was more pronounced for the D614G Spike (3.55-fold increase) comparatively to the Spikes from the variants tested (Beta, Delta, Mu and A.2.5) (1.76–1.99-fold increase). While the Delta, Mu and A.2.5 Spikes displayed increased ACE2 binding compared to the D614G Spike at 37 °C, this phenotype was lost at 4 °C. As previously described [24], the B.1.351 Spike exhibited an increased ACE2 binding at both 37 °C and 4 °C.

To further investigate the effect of temperature on ACE2 interaction, we assessed the binding kinetics of selected RBD single mutants to soluble ACE2 (sACE2) by biolayer interferometry (BLI). As previously described, biosensors were coated with RBD proteins and incubated with increasing concentrations of sACE2, ranging from 31.25 to 500 nM [22]. We observed that binding affinity was increased for all the indicated RBD mutants at lower temperature (Figure 2A–C, Appendix A). This higher affinity could be explained by a major decrease in the off-rate kinetics at 10 °C compared to 25 °C for all RBD mutants, regardless of differences in on rate kinetics (Figure 2A–C, Appendix A). Among the single mutations present in the tested variants, only the N501Y mutation (present in both B.1.351 and B.1.621) enhanced the affinity of RBD for ACE2. All the other RBDs required low temperature to reach this level of affinity. As previously reported, the RBD N501Y for ACE2 exhibited an enhanced affinity for ACE2 at both 10 °C and 25 °C (21). In contrast, introduction of the K417N mutation (present in the B.1.351) into RBD reduced affinity for ACE2 at both temperatures. Altogether, our results indicate that lower temperatures facilitate ACE2–RBD interaction.

### 3.2. Recognition and Neutralization of Spike Variants by Plasma from Vaccinated Individuals

To assess the antigenic profile of these Spikes, we used plasma collected three weeks after the second dose, with a 16 week interval between doses [30], of the BNT162b2 vaccine in naïve or previously infected individuals (Figure 3A,B, Appendix A). As described above, HEK 293T cells were transfected with Spike variants, incubated with the indicated plasmas followed by flow cytometry analysis, as described in several other studies [25,27,28,31,32]. All the SARS-CoV-2 S variants were recognized less efficiently compared to D614G S with the Delta S presenting the most profound decrease in plasma recognition.

We then determined the neutralization profile of the different emerging Spike variants using a pseudoviral neutralization assay [27,28,31,32,33]. Serial dilutions of plasma from vaccinated individuals were incubated with pseudoviruses bearing the different Spike variants before incubation with HEK 293T cells stably expressing the ACE2 receptor [25,27]. We observed a significant reduction in the capacity of plasma from naïve and previously-infected vaccinated individuals to neutralize pseudoviral particles bearing the Beta, Delta and Mu variants (Figure 3C,D, Appendix A).

## 4. Discussion

In this study, we analyzed the Spike glycoproteins of emerging variants Mu and A.2.5 and compared them to the Beta and Delta Spike variants. In line with previous reports, Beta and Delta Spikes showed a significant increase in ACE2-Fc binding [24,34,35,36], Mu and A.2.5 also bound ACE2 better than D614G but to a lower extent than Beta and Delta Spikes. Whether this differential ACE2 interaction influences the transmission rate of these variants remains unclear.

We previously reported that temperature variation impacts the RBD–ACE2 interface by modulating the Spike trimer conformation [22] and several VOCs/VOIs showed significant increase in ACE2 binding at cold temperature (4 °C) [24]. Accordingly, we observed that both Mu and A.2.5 variants presented better ACE2 interaction at lower temperatures. Further, RBD mutants of these indicated SARS-CoV-2 variants interacted with increased affinity to the ACE2 receptor at lower temperature suggesting that lower temperature promotes the thermodynamic stability of the ACE2–Spike–RBD complex whereas some of the mutations, such as N501Y, bypass this requirement. Thus, lower temperature and critical mutations may impart increased viral transmissibility and replication by modulating the Spike–ACE2 interaction. We previously demonstrated that the N501Y mutation significantly impacts ACE2 interaction independently of the temperature [22,24]. Accordingly, variants harboring the N501Y mutation, such as the Alpha, Beta and Gamma, displayed improved ACE2 interaction compared to the D614G Spike at cold temperature, revealing the crucial role of this mutation in facilitating Spike–ACE2 interaction. However, despite harboring this mutation (N501Y), the capacity of the Mu Spike to favor ACE2 binding over the D614G was not observed at cold temperature. This suggests that other changes present in this Spike also impact ACE2 interaction at cold temperature.

Finally, we found that Mu and A.2.5 Spikes were recognized by plasma from vaccinated naïve and previously-infected individuals, albeit to a lesser extent than D614G S. This lower recognition translated into lower neutralization capacity for Mu but not for A.2.5, suggesting that binding to the Spike does not necessarily translate into neutralization, in agreement with previous observations [33]. In summary, our results highlight the importance of measuring critical parameters, such as ACE2 interaction, plasma recognition and neutralization, from each emerging variant.

## Figures and Tables

**Figure 1 viruses-14-00144-f001:**
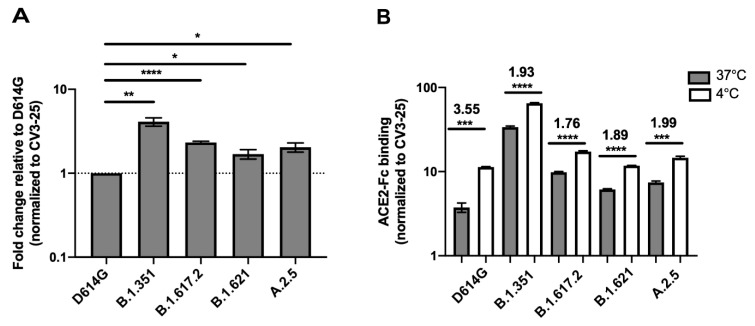
Spike glycoproteins interaction with ACE2. HEK 293T cells were transfected with the indicated SARS-CoV-2 Spike variants. 48 h post transfection, cells were stained with ACE2-Fc or with CV3-25 Ab and analyzed by flow cytometry. ACE2-Fc binding to the different full Spike variants is presented as a ratio of ACE2 binding of D614G Spike (**A**). The graph represented ACE2-Fc binding to the different full Spike variants at 37 °C and 4 °C (**B**). For each Spike variant, statistical significance and fold changes of ACE2 binding at 4 °C vs. 37 °C is indicated in (**B**). ACE2-Fc binding was normalized to CV3-25 binding in each experiment and at each indicated temperature. Error bars indicate means ± SEM. Statistical significance was performed using Mann–Whitney U test (* *p* < 0.05; ** *p* < 0.01; *** *p* < 0.001; **** *p* < 0.0001).

**Figure 2 viruses-14-00144-f002:**
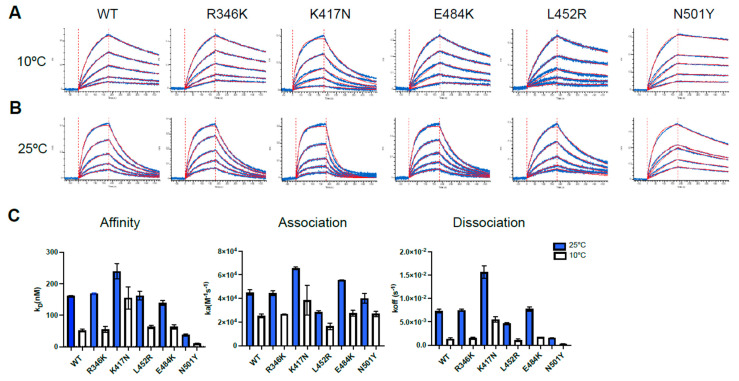
Analysis of hACE2-RBD binding affinity by Biolayer Interferometry. (**A**–**C**) Binding kinetics between SARS-CoV-2 RBD (WT or different single mutant) assessed by BLI at two different temperatures. Biosensors coated with RBD proteins were incubated in two-fold dilution series of sACE2 (500 nM–31.25 nM) at 10 °C (**A**) and 25 °C (**B**) temperatures. Representative raw data are shown in blue and fitting model is shown in red. Graphs represent the affinity constants (K_D_), on rates (K_a_) and off rates (K_dis_), (**C**) values obtained in two different experiments at two different temperatures and calculated using a 1:1 binding model. All BLI data are summarized in Appendix A.

**Figure 3 viruses-14-00144-f003:**
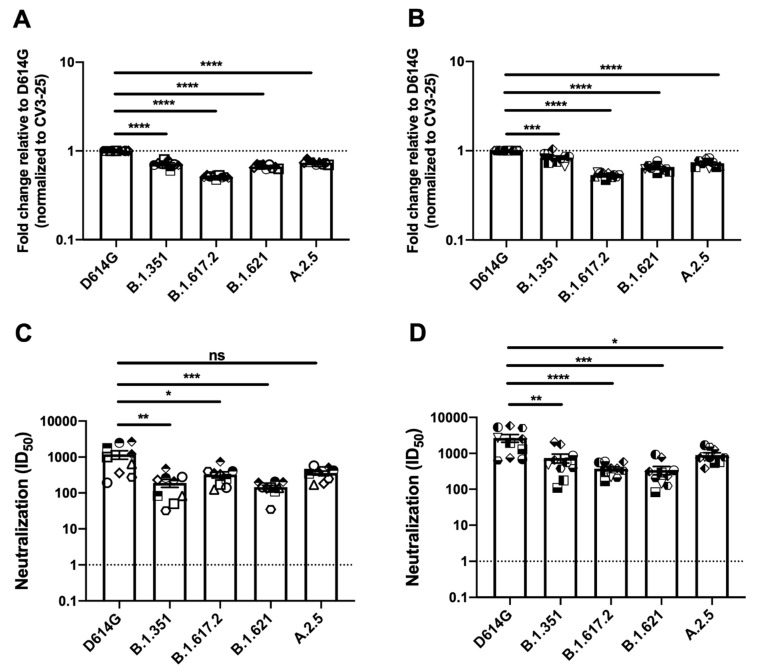
Evaluation of plasma recognition and neutralization ability of SARS-CoV-2 Spike variants of naïve or previously infected vaccinated individuals. HEK 293T cells were transfected with the indicated SARS-CoV-2 Spike variants. Two days post transfection, cells were stained with ACE2-Fc, 1:250 diluted plasma collected from naive vaccinated (*n* = 9) or previously infected individuals (*n* = 10) for each group or with CV3-25 Ab as control and analyzed by flow cytometry. Plasma recognition of second dose vaccinated naïve individuals (**A**) and previously infected second dose vaccinated individuals (**B**) are presented as ratio of plasma binding to D614G Spike normalized CV3-25 binding. Neutralizing activity of same group of individuals against pseudoviruses bearing the SARS-CoV-2 Spike variants were assessed. Pseudoviruses with serial dilutions of plasma were incubated for 1 h at 37 °C before infecting 293T-ACE2 cells. ID_50_ against pseudoviruses were calculated by a normalized non-linear regression using GraphPad Prism software. Neutralization activity of second dose vaccinated naïve individuals (**C**), and previously infected second dose vaccinated individuals (**D**) are represented. Limits of detection were indicated with a dotted line in the graph (ID_50_ = 30). Error bars indicate means ± SEM. Statistical significance was performed using Mann–Whitney U test (* *p* < 0.05; ** *p* < 0.01; *** *p* < 0.001; **** *p* < 0.0001, ns, non-significant).

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
