# Peer review of "Antigenicity of the Mu (B.1.621) and A.2.5 SARS-CoV-2 Spikes"

_viruses, 2022, doi:10.3390/v14010144_

Round 1

Reviewer 1 Report

This paper by Finzi and coworkers is well written and clear and addresses a limited set of questions on the antigenicity of the Mu and A.2.5 SARS CoV-2 spikes.  The authors address the temperature dependence of ACE2 binding of Mu and A.2.5 relative to other variants, as well as binding and neutralization by plasma from vaccinated individuals.  All spikes tested bound better to ACE2 compared to D614G spike with increased binding at 4C. Mu and A.2.5 share mutations with other variants, and single-mutant RBDs were also assessed for ACE2 binding. Only the N501Y mutant enhance affinity for ACE2.  For plasma binding assessment, spikes expressed on the surface of HEK293 cells were exposed to vaccinee plasma, and all variants were recognized less efficiently than D614G.  For neutralization, vaccine sera neutralization of beta, delta, and mu variants was significantly diminished. 

Overall this is a limited study, but well performed and clear. 

I have only minor comments:

  • Line 201: “the lower affinity…” should read “the higher affinity…”
  • Line 250: “extend” should be “extent”
  • Line 261: “increase” should be “increased”

Author Response

Dear Editor,

We were pleased by the reviews of our manuscript viruses-1529941 “Antigenicity of the Mu (B.1.621) and A.2.5 SARS-COV-2 Spikes”. Both reviewers agreed that the manuscript was well written, the study was well conducted, the results convincing, and the conclusions supported by the results. We are happy to address the reviewers’ comments to help strengthen our manuscript. Please find below point-by-point response to the comments.

Reviewer #1 (Comments for authors):

This paper by Finzi and coworkers is well written and clear and addresses a limited set of questions on the antigenicity of the Mu and A.2.5 SARS CoV-2 spikes.  The authors address the temperature dependence of ACE2 binding of Mu and A.2.5 relative to other variants, as well as binding and neutralization by plasma from vaccinated individuals.  All spikes tested bound better to ACE2 compared to D614G spike with increased binding at 4C. Mu and A.2.5 share mutations with other variants, and single-mutant RBDs were also assessed for ACE2 binding. Only the N501Y mutant enhance affinity for ACE2.  For plasma binding assessment, spikes expressed on the surface of HEK293 cells were exposed to vaccinee plasma, and all variants were recognized less efficiently than D614G.  For neutralization, vaccine sera neutralization of beta, delta, and mu variants was significantly diminished. 

Overall this is a limited study, but well performed and clear. 

Response: We thank reviewer #1 for his/her very positive assessment of our manuscript

I have only minor comments:

Line 201: “the lower affinity…” should read “the higher affinity…”

Response: thank you for pointing this out.  It is now corrected.

Line 250: “extend” should be “extent”

Response: this was corrected.

Line 261: “increase” should be “increased”

Response: this was corrected.

Reviewer #2 (comments for authors):

There are some minor typos:

  1. Lines 95, 106, CO2 should be CO2

Response: done.

  1. Line 161, KD should be KD

Response: done.

There are some points that it would be to discuss:

  1. Lines 258-260. Here the authors mention that lowering the T promotes thermodynamic stability. This argument is weak. It would be better if they were more precise. The fact that the KD decreases with T suggest that binding has a large entropic barrier. Perhaps it would be better to discuss this point in more precise terms. I suggest this becuase from their results one could propose that mutations that decrease the entropic barrier of binding are more susceptible to scape the immune respone from previous vaccines or infections.

Response: we thank the reviewer for raising this valid point.  However, since we have no data suggesting that mutations decreasing the entropic barrier for ACE interaction are more susceptible to escape from immune responses, we prefer to keep it as it is: we suggest that it could affect transmissibility and replication by modulating Spike-ACE interaction.

  1.  Figure 2. Were these experiments done once? if so they need to be repeated. If there are multiple experiments please add the error bars.

Response: as requested by the reviewer, the experiment was repeated and error bars added.  Since results were almost identical to what we reported in the original submission, the interpretation of this figure remains the same and therefore no changes in the text were made.

We trust that, with these changes, the manuscript is now suitable for publication in Viruses.

Sincerely,

Andrés Finzi

Reviewer 2 Report

There are some minor typos:

  1. Lines 95, 106, CO2 should be CO2
  2. Line 161, KD should be KD

There are some points that it would be to discuss:

  1. Lines 258-260. Here the authors mention that lowering the T promotes thermodynamic stability. This argument is weak. It would be better if they were more precise. The fact that the KD decreases with T suggest that binding has a large entropic barrier. Perhaps it would be better to discuss this point in more precise terms. I suggest this becuase from their results one could propose that mutations that decrease the entropic barrier of binding are more susceptible to scape the immune respone from previous vaccines or infections.
  2.  Figure 2. Were these experiments done once? if so they need to be repeated. If there are multiple experiments please add the error bars.

Author Response

(The authors gave the same response as above.)
